# *Tm*Spz4 Plays an Important Role in Regulating the Production of Antimicrobial Peptides in Response to *Escherichia coli* and *Candida albicans* Infections

**DOI:** 10.3390/ijms21051878

**Published:** 2020-03-09

**Authors:** Tariku Tesfaye Edosa, Yong Hun Jo, Maryam Keshavarz, Young Min Bae, Dong Hyun Kim, Yong Seok Lee, Yeon Soo Han

**Affiliations:** 1Department of Applied Biology, Institute of Environmentally-Friendly Agriculture (IEFA), College of Agriculture and Life Sciences, Chonnam National University, Gwangju 61186, Korea; bunchk.2000@gmail.com (T.T.E.); yhun1228@jnu.ac.kr (Y.H.J.); mariakeshavarz1990@gmail.com (M.K.); ugisaka@naver.com (Y.M.B.); kjsdh3@hotmail.com (D.H.K.); 2Ethiopian Institute of Agricultural Research, Ambo Agricultural Research Center, Ambo 37, Ethiopia; 3Department of Life Science and Biotechnology, College of Natural Sciences, Soonchunhyang University, Asan city 31538, Korea; yslee@sch.ac.kr

**Keywords:** AMP expression, defense response, mealworm, Spätzle, Toll receptor

## Abstract

Spätzle family proteins activate the Toll pathway and induce antimicrobial peptide (AMP) production against microbial infections. However, the functional importance of *Tm*spätzle4 (*TmSpz4*) in the immune response of *Tenebrio molitor* has not been reported. Therefore, here, we have identified and functionally characterized the role of *TmSpz4* against bacterial and fungal infections. We showed that *TmSpz4* expression was significantly induced in hemocytes at 6 h post-injection with *Escherichia coli*, *Staphylococcus aureus*, and *Candida albicans*. *TmSpz4* knock-down significantly reduced larval survival against *E. coli* and *C. albicans.* To understand the reason for the survivability difference, the role of *TmSpz4* in AMP production was examined in *TmSpz4*-silenced larvae following microbe injection. The AMPs that are active against Gram-negative bacteria, including *TmTenecin-2*, *TmTenecin-4*, *TmAttacin-1a*, *TmDefensin-2*, and *TmCecropin-2*, were significantly downregulated in response to *E. coli* in *TmSpz4*-silenced larvae. Similarly, the expression of *TmTenecin-1*, *TmTenecin-3*, *TmThaumatin-like protein-1* and *-2*, *TmDefensin-1*, *TmDefensin-2*, and *TmCecropin-2* were downregulated in response to *C. albicans* in *TmSpz4*-silenced larvae. In addition, the transcription factor NF-κB (*TmDorX1* and *TmDorX2)* expression was significantly suppression in *TmSpz4*-silenced larvae. In conclusion, these results suggest that *TmSpz4* plays a key role in regulating immune responses of *T. molitor* against to *E. coli* and *C. albicans*.

## 1. Introduction

Insects are the largest and most diverse group of animals on Earth, and have highly adaptable defenses against different environmental threats, including microorganisms (bacteria, fungi, and viruses) and parasites. To survive against microorganisms, insects have developed a potent defense mechanism, known as innate immunity, which can recognize and eliminate microbes [1]. This well-developed system includes physical defenses, as well as cellular and humoral immunity. The humoral immune system principally relies on antimicrobial peptides (AMPs), lectins, lysozyme, and protease inhibitors [2]. The innate immune system depends on pattern recognition receptors (PRRs) that recognize conserved molecules on pathogens, called pathogen-associated molecular patterns [3]. Recognition of invading pathogens activates signal transduction pathways [Toll or immune deficiency (IMD)], leading to the expression of AMPs that attack the invading pathogens [4,5].

Since Toll was first identified as a regulator of dorsoventral axis establishment during embryonic development [6], scientists have made great progress in developing genetic and molecular biological methodologies to demonstrate the functional role of the Toll pathway in the *Drosophila* immune system [7,8,9]. Specifically, the Toll pathway is activated when peptidoglycan recognition proteins (PGRPs) or Gram-negative bacteria binding proteins (GNBPs) recognize microbe-derived peptidoglycan or β-1,3 glucan [10] and the signal generated by this recognition is conveyed to a proteolytic cascade that leads to cleavage of the cytokine-like protein pro-spätzle. The resulting mature spätzle functions as a ligand that activates the Toll receptor [11,12]. Spätzle is an extracellular cytokine-like protein and a Toll receptor ligand. Its inactive form, pro-spätzle, contains a signal peptide, a regulatory N-terminal pro-domain, and a signaling precursor consisting of a C-terminal active fragment of 106 amino acids [13,14]. Active spätzle is generated during development and/or during the immune response to microbial infection. More specifically, during an immune response, a proteolytic cascade cleaves pro-spätzle, releasing active spätzle [11].

In *Drosophila*, six spätzle homologues (*Spz1–6*) have been identified, including the first identified spätzle gene, *spz-1*, which encodes a protein containing neurotrophin-like cystine-knot domains [15]. The study of the interaction of spätzle family proteins with each other and with Toll receptors during the activation of AMP gene expression showed that the binding of Spz-1, -2, and -5 to the Toll-1 and Toll-7 ectodomains promotes the activation of *drosomycin* and several other AMP genes [16,17,18].

The functions of the spätzle family proteins have been investigated in *Anopheles gambiae* and *Aedes aegypti* in response to fungal challenge [19,20], in *Bombyx mori* in response to microbial infection [21], and in *Manduca sexta* in response to Gram-positive bacterial, Gram-negative bacterial, and fungal infections [22]. In *B. mori*, spätzle4 has been shown to play an important role against Gram-negative bacteria, Gram-positive bacteria, and fungi, specifically in the integument [23].

In shrimp (*Litopenaeus vannamei*), a spätzle gene (*LvSpz4*) was functionally characterized and shown to be involved in innate immunity; more specifically, it was shown to be involved in the cross talk between the TLR-NF-κB pathway and unfolded protein response (UPR) [24]. A spätzle-like protein was identified in Chinese shrimp (*Fenneropenaeus chinensis*) and shown to function in the innate immune responses to bacteria and viruses [25].

In the last decade understanding of *T. molitor* proteolytic cascade has increased greatly due to the intensive biochemical studies on modular serine protease (MSP), spätzle-processing enzyme (SPE)-activating enzyme (SAE), and SPE which results in pro-spätzle cleavage [11,26]. The importance of spätzle genes in innate immune responses against microbial infections extends from insects to shrimp. However, despite their biochemical characterization, the functions of spätzle genes in the *T. molitor* immune response to microbial challenge have remained elusive. In order to further investigate spätzle genes, we have conducted RNA-seq and genome sequencing, and thus identified nine spätzle genes (*TmSpz-like*, -*1b*, -*3*, -*4*, -*5*, -*6*, -*7*, -*7a*, and -*7b*). In the current study, we focused on the identification and functional characterization of *Tm*Spätzle4 in immune responses against Gram-positive and Gram-negative bacteria and fungi.

## 2. Results

### 2.1. Sequence Identification and Phylogenetic Analysis Of TmSpz4

The full-length cDNA sequence of *TmSpz4* was obtained from the *T. molitor* RNAseq database by a local tblastn search of the *T. molitor* nucleotide database using the *T. castaneum* spätzle4 protein sequence as the query. The *TmSpz4* open reading frame (ORF) is 1143 bp long, and it encodes a 380 amino acid long protein (Figure 1). The 5′- and 3′-untranslated regions (UTR) of *TmSpz4* were 203 and 820 bp in length, respectively. Domain analysis suggested that *Tm*Spz4 contains one cystine-knot domain at the C-terminus, which is a ligand of the Toll receptor; one cleavage site that is predicted to be processed by SPE (Figure 1); and a predicted signal peptide. Phylogenetic analysis revealed that the Spz4 sequences of the Coleopteran insects (including *Tribolium castaneum* spätzle 4) were grouped together (Appendix A).

### 2.2. Developmental and Tissue-Specific Expression Patterns of TmSpz4

The expression patterns of *TmSpz4* mRNA transcripts in mealworm across developmental stages and tissues were examined by RT-qPCR. *TmSpz4* transcript expression was observed in all analyzed developmental stages and tissues, and the highest expression was observed at the prepupal and 4-day-old pupal stages. The mRNA levels increased from the young larval stage to the prepupal stage and from the 1-day-old pupal stage to the 4-day-old pupal stage (Figure 2A). In pupae, once expression peaked, it gradually decreased through the rest of the pupal stages. In adults, *TmSpz4* expression was constantly low, except in 1-day-old adults, in which it was slightly higher.

Examination of expression levels in different tissues revealed that *TmSpz4* was highly expressed in the hemocytes of late larvae (Figure 2B), while in adults, *TmSpz4* expression was highest in the integument, followed by the hemocytes, fat body, ovaries, and testes (Figure 2C). Conversely, *TmSpz4* expression levels were low in the integument and Malpighian tubules of late larvae and in the gut and Malpighian tubules of adults.

### 2.3. Temporal Induction Pattern of TmSpz4

To determine whether *TmSpz4* expression is induced by immune challenge, the expression of *TmSpz4* in *T. molitor* larvae was examined over time after injecting *E. coli*, *S. aureus*, or *C. albicans*. PBS (pH 7) was injected as a control. Then, three immune tissues (hemocytes, fat body, and gut) were collected at 3, 6, 9, 12, and 24 h post-injection to isolate total RNA, and *TmSpz4* expression in them was analyzed by RT-qPCR. Microbial challenge time-dependently induced the transcription of *TmSpz4* in all tested tissues. The highest expression was observed in hemocytes at 6 h post-infection of all test microorganisms (Figure 3A). In the gut, injection of *E. coli* and *S. aureus* highly induced *TmSpz4* expression at 9 and 24 h post-injection, respectively (Figure 3C), whereas in the fat body, the highest expression was detected at 24 h post-injection of *E. coli* and *C. albicans* (Figure 3B).

### 2.4. Effect of TmSpz4 RNAi on T. Molitor Survivability

Based on the observed temporal induction of *TmSpz4* following microorganism injection, we sought to determine the role of *TmSpz4* in resistance to bacteria and fungi by silencing *TmSpz4* transcript levels in *T. molitor* larvae through RNAi. *TmSpz4* mRNA levels were decreased by 90% 5 days after ds*TmSpz4* injection (Figure 4A).

After confirming the efficient knock-down of *TmSpz4* in larvae, they were challenged with bacteria or fungi. The survival of *TmSpz4*-silenced *T. molitor* larvae following microbial injection was monitored for 10 days. Injection of ds*TmSpz4* and/or ds*EGFP* did not affect the survival of PBS*-*injected *T. molitor* larvae. However, ds*TmSpz4-*injected larvae were significantly more susceptible to *E. coli* (64.7%) (Figure 4B) and *C. albicans* (47%; Figure 4D). In contrast, the survival rates of ds*TmSpz4-*injected larvae did not differ significantly from that of the control after infection with *S. aureus* (Figure 4C).

### 2.5. Effects of TmSpz4 Gene Silencing on the Expression of AMPs

The survival study showed that *TmSpz4* knock-down reduced the survival of *T. molitor* larvae following challenge with *E. coli* and *C. albicans*, suggesting the importance of *TmSpz4* in the immune defense against Gram-negative bacteria and fungi. Thus, to characterize the function of TmSpz4 in the production of AMPs in response to microbial infection, *TmSpz4* expression was silenced in *T. molitor* larvae, and the larvae were challenged with *E. coli*, *S. aureus*, or *C. albicans*. Then, the expression levels of 14 different AMP genes were assessed at 24 h post-infection.

In hemocytes (Figure 5), the expression levels of the AMP genes were significantly reduced in *TmSpz4*-silenced larvae following microbial challenge, including that of *TmTen-2* by *E. coli*; those of *TmTen-2*, *-3*, *-4*, and *TmCec-2* by *S. aureus*; and those of *TmTen-1*, *-2*, *-3*, *TmAtt-2*, *TmCol-2*, *TmTLP-1*, and *TmTLP-2* by *C. albicans*. In the fat body, the expression levels of *TmTen-2*, *-3*, *-4*, *TmAtt-1a*, *TmDef-2*, *TmTLP-2*, and *TmCec-2* were reduced by *E. coli*; those of *TmTLP-1* and *TmCec-2* were reduced by *S. aureus*; and those of *TmTen-1*, *-2*, *-4*, *TmAtt-1a*, *TmAtt-1b*, *TmAtt-2*, *TmDef-1*, *TmDef-2*, *TmTLP-1*, and *TmTLP-2* were reduced by *C. albicans* in *TmSpz4*-silenced larvae (Figure 6). In the gut of *TmSpz4*-silenced larvae, the expression levels of *TmTen-4*, *TmAtt-1a*, and *TmCec-2* were reduced by *E. coli*; those of *TmTen-3*, *-4*, *TmDef-2*, and *TmCec-2* were reduced by *S. aureus*; and those of *TmTen-2*, *-4*, *TmAtt-2*, *TmDef-1*, *TmTLP-1*, and *TmCol-2* were reduced by *C. albicans* when compared with the levels in ds*EGFP*-injected larvae (Figure 7).

Interestingly, in contrast, *TmSpz4* knock-down increased the mRNA levels of *TmTen-4*, *TmAtt-1a*, *TmAtt-1b*, *TmCol-1*, *TmCol-2*, and *TmDef-2* in the hemocytes of *E. coli-*challenged larvae (Figure 6). Similarly, the expression levels of *TmAtt-1b*, *TmAtt-2*, and *TmCol-1* in the fat body (Figure 6) and those of *TmTen-2*, *TmAtt-1b*, *TmAtt-2*, *TmCol-1*, and *TmDef-2* in the gut (Figure 7) were increased in *TmSpz4-*silenced larvae after *E. coli* injection.

The expression of NF-κB genes, *TmDorX1*, *TmDorX2*, and *TmRelish*, were investigated under the same conditions as those used in the AMP expression experiment. ds*TmSpz4* injection significantly decreased the expression levels of *TmDorX1* and *TmDorX2* in the fat body following challenge with *E. coli* (Figure 8A,B). Similarly, *TmSpz4* knock-down significantly reduced the expression of *TmDorX1* in hemocytes and *TmDorX2* in the gut following *C. albicans* challenge (Figure 8A,B). In the gut, injection of *TmSpz4* RNAi upregulated the expression of *TmRelish* following challenge with all test microorganisms (Figure 8C).

## 3. Discussion

The Toll receptor, which plays important roles in the production of AMPs in response to infection in insects, is activated by the endogenous cytokine ligand spätzle [14,27,28]. Thus, the molecular functions of spätzle proteins in Toll receptor activation and the subsequent activation of AMPs have been well studied in various insects. For example, in *Drosophila*, Spz-1, -2, and -5 bind to Toll-1 and Toll-7 to produce drosomycin and several other AMPs [16,29]. Additionally, in *B. mori*, spätzle-1 binds to the Toll receptor to activate the production of attacin-1, cecropin-6, and moricin [30].

In current study, *TmSpz4* of *T. molitor* was identified and functionally characterized. The identified *Tm*Spz4 protein contains one cystine-knot domain at the C-terminus, which is a Toll receptor ligand; one cleavage site, which is predicted to be processed by SPE; and one signal peptide, which enables its transport through cell membranes. Spätzle is synthesized in an inactive form (pro-spätzle), the N-terminus signal peptide of which is removed to allow the secretion of mature spätzle [27].

A previous study on *Drosophila* demonstrated a cross talk between a steroid hormone (ecdysone) or juvenile hormone and immune-related genes. Briefly, the ecdysone hormone activated a nuclear receptor to generate a heterodimer with ultraspiracle, promoting the transcription of immune-related genes [31]. It is well known that ecdysone is critical for the activation of AMP gene expression and phagocytosis [32]. Investigation of the expression of ecdysone during the development of *Drosophila* by radioimmune assay showed that ecdysone activity was highest during the pupal, prepupal, and late larval stages (in descending order) [33]. Accordingly, in our study as well, developmental stage and tissue-specific expression analysis of *TmSpz4* revealed the highest expression during the prepupal and pupal stages, in the hemocytes of larvae, and in the integument, hemocytes, and fat body of adults. Given that hemocytes play important roles in immunity, nutrient transportation, and growth hormone synthesis [34,35]. Taken together, these data shows that the highest *TmSpz4* expression is during transitional stages and in growth hormone-synthesizing tissues (hemocytes) under normal conditions. The molecular relationship between developmental hormone and *TmSpz4* during normal developmental condition needs to be investigated to further understand the role of developmental hormones in the expression of *TmSpz4*. The Toll receptor ligand spätzle was activated when PGRPs or GNBP recognized peptidoglycan (PGN) or β-1,3 glucan from Gram-positive bacteria or fungi [10,14]. Pattern recognition proteins (e.g., PGRPs and GNBP) are found on the plasma membrane of fat body and hemocytes cells [36]. Consequently, in the current study, the highest and earliest induction of *TmSpz4* in *T. molitor* larvae following challenge with *S. aureus*, *C. albicans*, and *E. coli* was observed in hemocytes. During infection in *Drosophila*, hemocytes synthesized and secreted signals that could be detected by the fat body [37], suggesting early recognition of infection by hemocytes. The induction of *TmSpz4* expression by *E. coli* suggests the presence of a signaling cross talk between the Toll and IMD pathways in *T. molitor*. Similarly, the previous in vitro experiments showed that *E. coli* induced the activation of spätzle in *T. molitor* larvae, suggesting that the polymeric DAP-type PGN forms a complex with *Tenebrio* PGRP-SA to activate the Toll receptor ligand, spätzle [30]. Similarly, in our recent publication we have reported *TmSpz6* is important in regulating the AMPs production in response to *E. coli* [38] The survival results in our study support the induction analysis results, implying the importance of *TmSpz4* in the defense response of *T. molitor* larvae against *C. albicans* and *E. coli* by regulating different AMPs production. Specifically, *TmSpz4* silencing resulted in increased susceptibility of the larvae to *E. coli* and *C. albicans* infections. Similarly, *TmSpz4* silencing suppressed the induction of several AMP genes following challenge with *E. coli* (*TmTen-2*, *TmTen-4*, *TmAtt-1a*, *TmDef-2*, and *TmCec-2*) and *C. albicans* (*TmTen-1*, *TmTen-3*, *TmTLP-1*, *TmTLP-2*, *TmDef-1*, *TmDef-2*, and *TmCec-*2). These suppressed AMPs have been shown to have antibacterial activity against both gram-negative bacteria and fungi [39,40,41,42]. In particular, glycine-rich AMPs, such as attacins, in *Hyalophora cecropia* [43] and *T. molitor* [44], as well as tenecin-2 and -4 in *T. molitor* [30,45] have been shown to be particularly effective against Gram-negative bacteria. The effectiveness of tenecin-3 [46], tenecin-1, tenecin-2 [47], and defensins [48] against fungal infections has also been reported.

Activation of either the Toll pathway by Lys-type PGN or β-1,3 glucan or the IMD pathway by DAP-type PGN leads to translocation of the NF-κB family transcription factors Dorsal and Relish [49]. The extracellular protein spätzle, which is generated during development and/or in response to microbial infection during the insect immune response, activates the Toll pathway [11]. Therefore, we wanted to determine if *Tm*Spz4 affects the expression of Dorsal and subsequent AMP production in *T. molitor* in response to microbial challenge. Thus, the transcription levels of Dorsal and Relish were quantified in *TmSpz4-*silenced *T. molitor* larvae challenged with different microbes. In agreement with the AMP expression results in this study, *TmDorX1* and *TmDorX2* were significantly suppressed in the hemocytes and fat body of *TmSpz4*-silenced larvae following challenge with *E. coli* or *C. albicans*, indicating that *Tm*Spz4 is important in the expression of *TmDorX2*. Active spätzle binds to Toll receptor to activate the Toll pathway; then, the MyD88-Tube-Pelle complex leads to the phosphorylation and degradation of Cactus, an inhibitor of NF-κB. Thus, after cactus is degraded, both Dorsal and Dif are translocated to the nucleus and bind to the κB-related sequences in AMP genes [50]. Taken together, the results of this study suggest the importance of *TmSp4* in the humoral immunity of *T. molitor* through the activation of the Toll pathway, subsequent activation of the NF-κB transcription factor (*TmDorX1* and *TmDorX2*), and ultimate production of AMPs against *E. coli* and *C. albicans* infection.

## 4. Materials and Methods

### 4.1. Insect Culture

The coleopteran insect *T. molitor* (commonly known as mealworm) was maintained at 27 ± 1 °C and 60 ± 5% relative humidity in the dark on an artificial diet containing 170 g whole-wheat flour, 20 g fried bean powder, 10 g soy protein, 100 g of wheat bran, 200 mL sterile water, 0.5 g chloramphenicol, 0.5 g sorbic acid, and 0.5 mL propionic acid. For the experiments, 10th–12th instar larvae were used. To ensure uniformity in size, the larvae were separated according to physical size using a set of laboratory test sieves (Pascall Eng. Co., Ltd., Crawley, UK).

### 4.2. Preparation of Microorganisms

A Gram-negative bacterial strain (*Escherichia coli* K12), a Gram-positive bacterial strain (*Staphylococcus aureus* RN4220), and a fungus (*Candida albicans* AUMC 13529) were used to study the function of *Tm*Spz4 in the innate immune response of mealworms against microbial infections. These microorganisms were cultured in Luria-Bertani (LB; *E. coli* and *S. aureus*) and Sabouraud dextrose (*C. albicans*) broths at 37 °C overnight and then subcultured at 37 °C for 3 h. Then, the microorganisms were harvested and washed twice with phosphate-buffered saline (PBS; pH 7.0) by centrifugation at 3500 rpm for 10 min. The microbes were suspended in PBS, and the cell density was determined by measuring the OD_600_. Finally, 106 cells/μL of *E. coli* and *S. aureus* and 5 × 104 cells/μL of *C. albicans* were separately prepared for use in the subsequent challenge experiments.

### 4.3. Identification and Cloning of Full-Length cDNA Sequence of TmSpz4

The *T. molitor TmSpz4* gene was identified by a local BLAST analysis using the amino acid sequence of the *T. castaneum spz4* gene (EFA09263.2) as the query. The partial cDNA sequence of *Tmspz4* was obtained from the *T. molitor* RNAseq database, and the full-length cDNA sequence of *TmSpz4* (MT075617) was identified by 5′- and 3′-rapid amplification of cDNA end (RACE) PCR using the SMARTer RACE cDNA amplification kit (Clontech, Mountain View, CA, USA), according to the manufacturer’s instructions. PCR was performed using the AccuPower^®^ PyroHotStart Taq PCR PreMix (Bioneer, Daejeon, Korea) and *TmSpz4*-specific primers (RACE primers: *TmAtg4*-cloning_Fw and *TmSpz4*-cloning_Rv; Table 1) under the following cycling conditions: pre-denaturation at 95 °C for 5 min, followed by 35 cycles of denaturation at 95 °C for 30 s, annealing at 53 °C for 30 s, and extension at 72 °C for 2 min, with a final extension step at 72 °C for 5 min on a MyGenie96 Thermal Block (Bioneer). The PCR products were purified using the AccuPrep^®^ PCR Purification Kit (Bioneer), immediately ligated into T-Blunt vectors (Solgent, Daejeon, Korea), and transformed into *E. coli* DH5α competent cells, according to the manufacturer’s instructions. Plasmid DNA was extracted from the transformed cells using the AccuPrep^®^ Nano-Plus Plasmid Extraction Kit (Bioneer) and then sequenced and analyzed. Finally, the full-length cDNA sequence of *TmSpz4* was obtained (Appendix A).

### 4.4. Domain and Phylogenetic Analysis

The domains of *Tm*Spz4 were analyzed using the InterProScan 5 and BLAST programs. A multiple sequence alignment was performed with representative Spz4 protein sequences from other insects obtained from GenBank using ClustalX2. Phylogenetic analyses of *Tm*Spz4 homologues protein were performed using the Clustal X2 and the phylogenic tree was constructed by MEGA7 programs using the maximum likelihood and bootstrapped of 1000 replications. The following Protein sequences were used to construct the phylogenetic tree. *Dm*Spz4; *Drosophila melanogaster spatzle 4* (AAF53100.2), *Dm*Spz6; *Drosophila melanogaster* spatzle 6 (AAF47261.1), *Dm*Spz5; *Drosophila melanogaster* spatzle 5 (AAF47694.1), *Dm*Spz; *Drosophila melanogaster* spatzle (AAF82745.1), *Dm*Spz3; *Drosophila melanogaster* spatzle 3 (AAF52574.2), *Dm*NTP1-H; *Drosophila melanogaster* neurotrophin 1, isoform H (AGB94113.1), *Dm*NTP1-E; *Drosophila melanogaster* neurotrophin 1, isoform E (ACZ94621.1), *Dm*NTP1-D; *Drosophila melanogaster* neurotrophin 1, isoform D (NP_001163348.1), *Tc*Spz7; *Tribolium castaneum* spatzle 7 (EEZ99267.2), *Tc*Spz4; *Tribolium castaneum* spatzle 4 (EFA09263.2), *Tc*Spz5; *Tribolium castaneum* spatzle 5 (EEZ97725.1), *Tc*Spz3; *Tribolium castaneum* spaetzle 3 (NP_001153625.1), *Tc*Spz6; *Tribolium castaneum* spatzle 6 precursor (NP_001164082.1), *Tc*Spz; *Tribolium castaneum* spaetzle (EEZ99207.1), *Tc*P-Spz; *Tribolium castaneum* PREDICTED: protein spaetzle (XP_008201191.1), *Tc*Spz-like; *Tribolium castaneum* spaetzle-like protein (EEZ99268.2).

### 4.5. TmSpz4 Expression and Temporal Induction Pattern Analysis

Total RNA was extracted from whole *T. molitor* (*n* = 20) at various developmental stages, including egg (EG), young instar larval (YL; 10th–12th instar larvae), late instar larval (LL; 19th–20th instar larvae), prepupal (PP), 1- to 7-day-old pupal (P1–7), and 1- to 5-day-old adult (A1–5) stages. To investigate tissue-specific *TmSpz4* expression, RNA was extracted from various tissues (*n* = 20), including the gut, hemocytes, integument, Malpighian tubules, and fat body of late instar larvae and the ovaries and testes of adults. To study the induction patterns of *TmSpz4* in different *T. molitor* larval tissues in response to microbial challenge, *E. coli* (Gram-negative bacteria), *S. aureus* (Gram-positive bacteria), or *C. albicans* (fungi) were injected into young instar larvae. Three immune-response related tissues, hemocytes, fat body, and gut, were collected at 3, 6, 9, 12, and 24 h post-injection into 500 μL of guanidine thiocyanate RNA lysis buffer (2 mL 0.5 M EDTA, 1 mL 1 M MES Buffer, 17.72 g guanidine thiocyanate, 0.58 g sodium chloride, 0.7 mg phenol red, 25 μL Tween-80, 250 μL acetic acid glacial, and 500 μL isoamyl alcohol) and homogenized using a homogenizer (Bertin Technologies, Montigny-le-Bretonneux, France) at 7500 rpm for 20 s.

Total RNA was extracted from the collected samples using the modified LogSpin RNA isolation method [51]. Briefly, homogenized samples were centrifuged for 5 min at 13,000 rpm and 4 °C. The diluted supernatant (300 μL) was transferred into a new 1.5 mL epitube, mixed with one volume of pure ethanol, transferred into a silica spin column (KA-0133-1; Bioneer, Daejeon, Korea), and centrifuged for 30 s at 13,000 rpm and 4 °C. The silica spin column was treated with DNase (M6101; Promega, WI, USA) at 25 °C for 15 min and washed with 3 M sodium acetate buffer and 80% ethanol. After drying by centrifugation at 13,000 rpm and 4 °C for 2 min, total RNA was eluted with 30 μL of distilled water (W4502-1L; Sigma-aldrich, MO, USA). The eluted RNA (2 μg) was used to generate cDNA using the AccuPower^®^ RT PreMix (Bioneer) and Oligo (dT) 12–18 primers on a MyGenie96 Thermal Block (Bioneer), according to the manufacturer’s instructions.

Quantitative PCR (qPCR) was performed using gene-specific primers, under the following cycling conditions: an initial denaturation step at 94 °C for 5 min, followed by 45 cycles of denaturation at 95 °C for 15 s and annealing at 60 °C for 30 s. The 2^−ΔΔCt^ method [52] was used to analyze *TmSpz4* expression levels. The *T. molitor* gene encoding ribosomal protein L27a (TmL27a) was used as an internal control for the normalization of differences in template concentration between samples.

### 4.6. Effect of TmSpz4 Gene Silencing in Response to Microorganisms

To synthesize the double-stranded RNA of the *TmSpz4* gene, forward and reverse primers containing the T7 promoter sequence at their 5′ ends were designed using the SnapDragon-Long dsRNA design software (https://www.flyrnai.org/cgibin/RNAi_find_primers.pl) (Table 1 and Appendix A). The PCR product was amplified using AccuPower^®^ Pfu PCR PreMix with *TmSpz4*_Fw and *TmSpz4*_Rv (Table 1) under the following cycling conditions: an initial denaturation step at 94 °C for 2 min, followed by 35 cycles of denaturation at 94 °C for 30 s, annealing at 53 °C for 30 s, and extension at 72 °C for 30 s, with a final extension step at 72 °C for 5 min. The PCR products were purified using the AccuPrep PCR Purification Kit (Bioneer), and dsRNA was synthesized using the Ampliscribe^TM^ T7-Flash^TM^ Transcription Kit (Epicentre Biotechnologies, Madison, WI, USA), according to the manufacturer’s instructions. After synthesis, the dsRNA was purified by precipitation with 5 M ammonium acetate and 80% ethanol. Subsequently, it was quantified using an Epoch spectrophotometer (BioTek Instruments, Inc., Winooski, VT, USA). As a control, the dsRNA of enhanced green fluorescent protein (ds*EGFP*) was synthesized, and all samples were stored at −20 °C until use.

Synthesized ds*TmSpz4* was diluted to a final concentration of 1 µg/µL and then injected into young-instar larvae (10th–12th instars; *n* = 30) using disposable needles mounted on a micro-applicator (Picospiritzer III Micro Dispense System; Parker Hannifin, Hollis, NH, USA). Another set of young-instar larvae (*n* = 30) were injected with equal amounts of ds*EGFP* as a negative control. Injected larvae were maintained on an artificial diet under standard rearing conditions. Evaluation of *TmSpz4* knock-down showed a greater than 90% reduction at 5 days post-injection.

To study the importance of *TmSpz4* in *T. molitor* immune responses, *TmSpz4*-silenced and ds*EGFP*-injected larvae were challenged with *E. coli* (10^6^ cells/μL), *S. aureus* (10^6^ cells/μL), or *C. albicans* (5 × 10^4^ cells/μL) in triplicate experiments. The challenged larvae were maintained for 10 days, and the number of living larvae were recorded during this time period. The survival rates of the *TmSpz4*-silenced group were compared to those of the control groups. Statistical analysis was conducted using the SAS 9.4 software (SAS Institute, Inc., Cary, NC, USA), and cumulative survival was analyzed by Tukey’s multiple test, at a significance level of *p* < 0.05.

### 4.7. Effect of TmSpz4 RNAi on AMP Expression against Microbial Challenge

To characterize the function of *TmSpz4* in the regulation of AMP gene expression in response to microbial infection, *TmSpz4* expression in larvae was first silenced using RNAi; then, these larvae were injected with *E. coli*, *S. aureus*, and *C. albicans*. Larvae injected with ds*EGFP* and PBS were used as the negative and injection controls, respectively. The hemocytes, fat body, and guts were dissected at 24 h post-microorganism injection. Total RNA was extracted and cDNA was synthesized as described above. Then, qRT-PCR was conducted using specific primers (Table 1) to analyze the temporal expression patterns of the fourteen AMP genes *TmTenecin-1* (*TmTen-1*), *TmTenecin-2* (*TmTen-2*), *TmTenecin-3* (*TmTen-3*), *TmTenecin-4* (*TmTen-4*), *TmAttacin-1a* (*TmAtt-1a*), *TmAttacin-1b* (*TmAtt-1b*), *TmAttacin-2* (*TmAtt-2*), *TmDefensin-1* (*TmDef-1*), *TmDefensin-2* (*TmDef-2*), *TmColptericin-1* (*TmCol-1*), *TmColptericin-2* (*TmCol-2*), *TmCecropin-2* (*TmCec-2*), *TmThaumatin-like protein-1* (*TmTLP-1*), and *TmThaumatin-like protein-2* (*TmTLP-2*).

### 4.8. Effects of dsTmSpz4 on the Expression Patterns of NF-κB Genes

To understand the effect of *TmSpz4*-RNAi on the expression of the NF-κB genes such as *TmDorsal-X1* (*TmDor-X1*), *TmDorsal-X2* (*TmDor-X2*), and *TmRelish* (*TmRel*), *TmSpz4* gene was silenced in the young instars larvae of *T. molitor* and other larval group were injected with ds*EGFP* as a negative control. Subsequently, *E. coli*, *S. aureus*, and *C. albicans*, were injected into *TmSpz4*-silenced and control larval group. After 24 h post microbial challenge, hemocytes, fat body and gut were dissected. Total RNA was extracted and cDNA was synthesized as described above. Then, qRT-PCR was conducted using *TmDorsal* and *TmRelish* specific primers (Table 1). All experiments were performed in triplicate.

### 4.9. Data Analysis

All experiments were triplicated. The mean expression of *TmSpz4* in the developmental stage were subjected to analysis of variance (ANOVA) using SAS 9.4 and means were compared by Tukey’s multiple range test (*p* < 0.05). Statistical analysis of survival analysis was carried out based on Kaplan-Meier plots (log-rank chi-square test; * *p* < 0.05). Comparative AMP gene expression was calculated using the delta delta Ct method (ΔΔCt). The fold change compared to the internal (*TmL27a*) and external (PBS) controls was calculated by the 2^–(ΔΔCt)^ method.

## Figures and Tables

**Figure 1 ijms-21-01878-f001:**
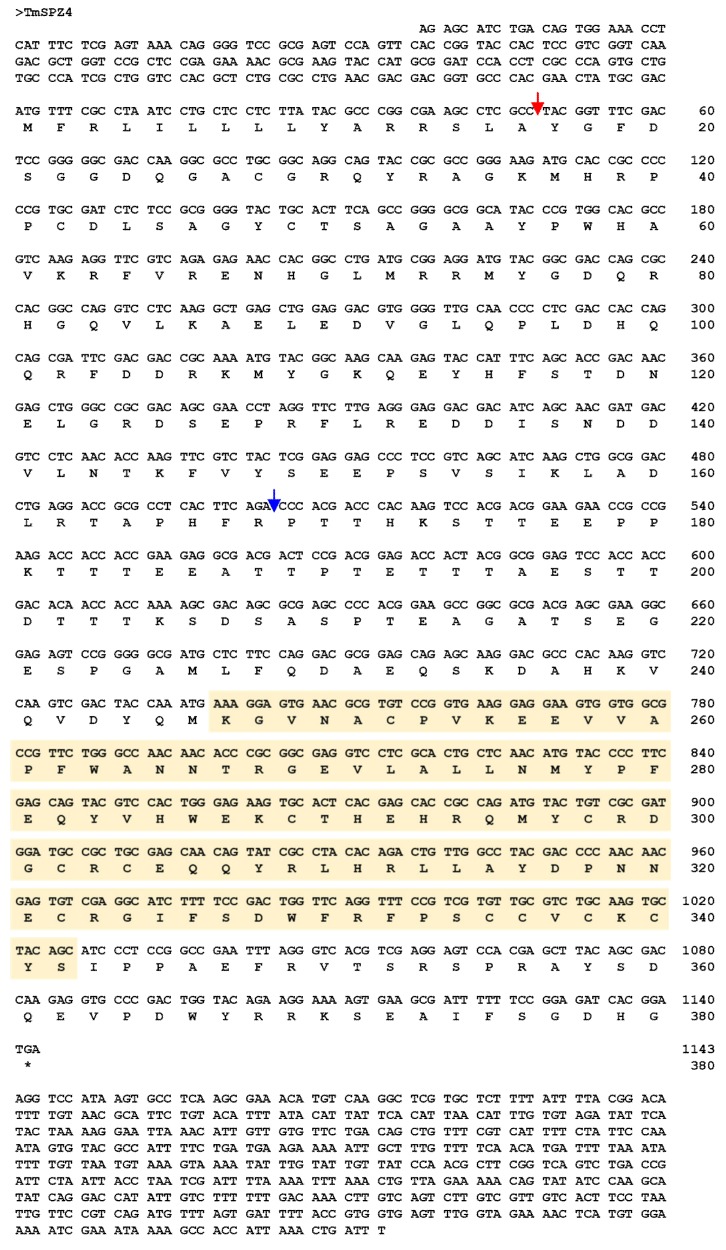
Nucleotide and deduced amino acid sequence of *TmSpz4. TmSpz4* contains a 1143 bp open reading frame encoding a predicted polypeptide of 380 amino acid residues. Domain analysis showed that *Tm*Spz4 includes one cystine-knot domain (yellow box), one signal peptide region (red arrow), and one cleavage site (blue arrow).

**Figure 2 ijms-21-01878-f002:**
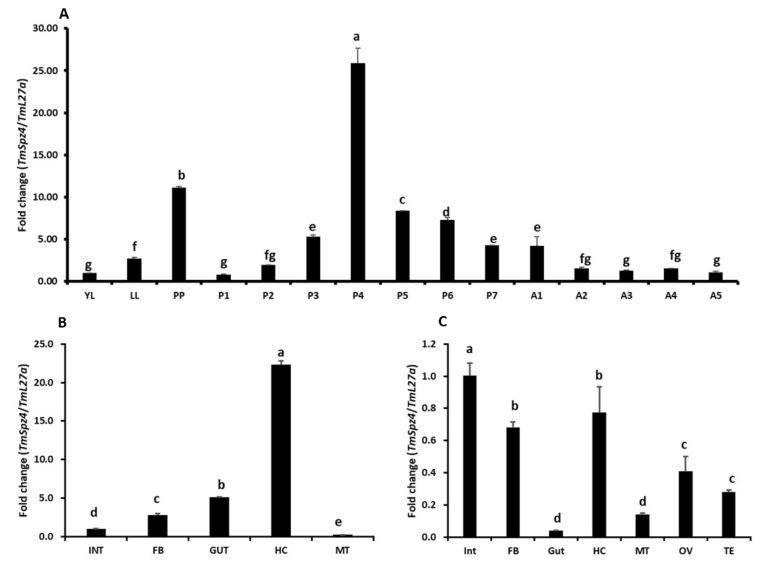
The expression patterns of *TmSpz4* gene in developmental and tissue of *T. molitor.* The developmental expression patterns of *TmSpz4* in mealworm at the young larval (YL), late larval (LL), pre-pupal (PP), 1–7-day-old pupal (P1–7), and 1–5-day-old adult (A1–5) stages were examined (**A**). In each experiment, RNA extracted from 20 individuals was used to synthesize cDNA. In larvae, *TmSpz4* expression gradually increased from the YL to the PP stage. In the pupae, the highest expression was observed at the 4-day-old pupal stage. In adults, there was no difference in *TmSpz4* expression from day 2 to day 5. Tissue-specific expression patterns of *TmSpz4* were also investigated in late larvae (**B**) and five-day-old adults (**C**). Hemocytes (HC), gut, fat body (FB), Malpighian tubules (MT), integument (INT) (for late instar larvae and adults), and testes (TE) and ovaries (OV) (for adults) were dissected and collected from 20 late larvae and 5-day-old adults. *T. molitor* 60S ribosomal protein L27a (*TmL27a*) was included as an endogenous control to normalize RNA levels among samples. The data are the means of three biological replicates. One-way ANOVA and Tukey’s multiple range test at 95% confidence level (*p* < 0.05) were performed and used to determine the level of significance of differences. The graph indicated by the same letter (a, b, c, d, e, f, g, fg) are not significantly different by Tukey’s multiple range (*p* < 0.05).

**Figure 3 ijms-21-01878-f003:**
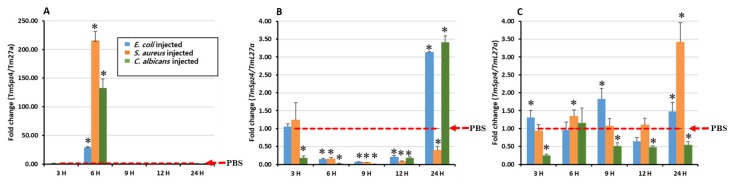
Induction patterns of *TmSpz4* in different *T. molitor* larval tissues. Temporal expression was analyzed in the hemocytes (**A**), fat body (**B**), and gut (**C**) of young larvae at 3, 6, 9, 12, and 24 h post-injection with *E. coli* (10^6^ cells/μL), *S. aureus* (10^6^ cells/μL), or *C. albicans* (5 × 10^4^ cells/μL). Twenty young mealworm larvae were used for each time point. *TmSpz4* expression levels were normalized to those in PBS-injected controls. *T. molitor* 60S ribosomal protein L27a (*TmL27a*) was used as an internal control. The dotted red line indicates PBS injection control. Asterisks indicate significant differences between infected and PBS injected larval group by Student’s *t*-test (*p* < 0.05). The vertical bars indicate Mean ± SD (*n* = 20).

**Figure 4 ijms-21-01878-f004:**
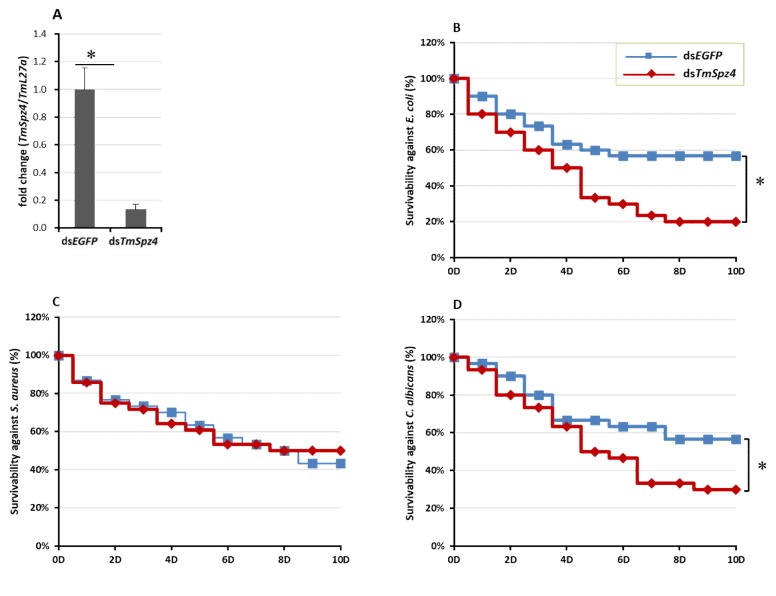
Effect of ds*TmSpz4* on the survival of *T. molitor* larvae. The silencing efficiency of dm*TmSpz4* was measured by qRT-PCR at 5 days post-injection (**A**). Then, the *TmSpz4*-silenced larvae were injected with *E. coli* (**B**), *S. aureus* (**C**), or *C. albicans* (**D**) and survival was monitored. ds*EGFP*-injected larvae were included as a negative control. The data are an average of three biologically independent replicate experiments. Asterisks indicate significant differences between ds*TmSpz4-* and ds*EGFP-*injected groups (*p* < 0.05). Statistical analysis of survival analysis was carried out based on Kaplan-Meier plots (log-rank chi-square test; * *p* < 0.05).

**Figure 5 ijms-21-01878-f005:**
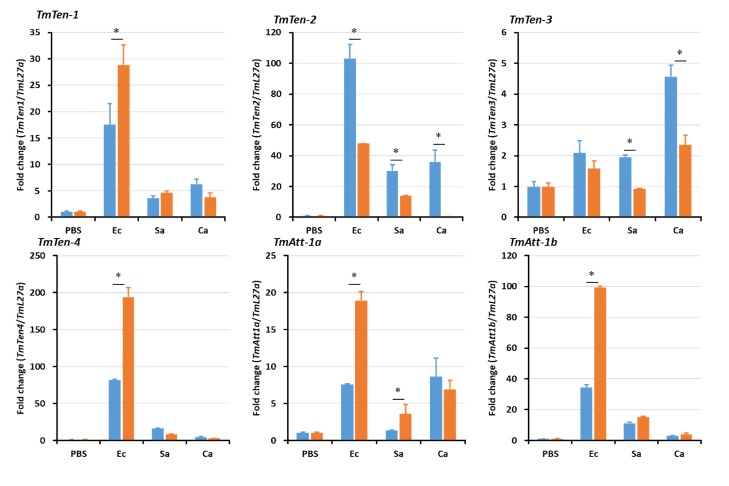
Antimicrobial peptide (AMP) induction patterns in hemocytes of *TmSpz4*-silenced larvae. *AMP* gene expression levels in the hemocytes of *TmSpz4*-knock-down *T. molitor* larvae were assessed after injection with *E. coli* (Ec), *S. aureus* (Sa), or *C. albicans* (Ca). PBS was injected as a control 5 days post-*TmSpz4* silencing. At 24 h post-microbial challenge, the expression levels of several AMP genes, including those of *TmTen-1*, *TmTen-2*, *TmTen-3*, *TmTen-4*, *TmAtt-1a*, *TmAtt-1b*, *TmAtt-2*, *TmDef-1*, *TmDef-2*, *TmCol-1*, *TmCol-2*, *TmCec-2*, *TmTLP-1*, and *TmTLP-2*, were measured by qRT-PCR. ds*EGFP* was injected as a negative control, and *TmL27a* expression was measured as an internal control. All experiments were performed in triplicate. Asterisks indicate significant differences between ds*TmSpz4-* and ds*EGFP-*treated groups when compared by Student’s *t*-test (*p* < 0.05).

**Figure 6 ijms-21-01878-f006:**
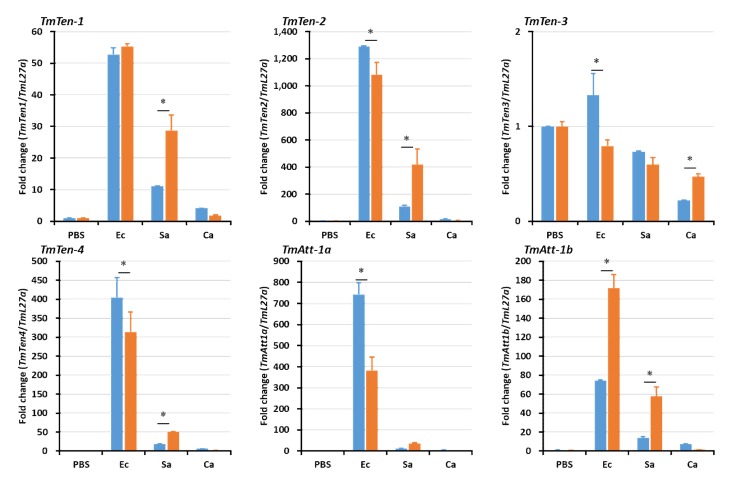
Antimicrobial peptide (AMP) induction patterns in fat body of *TmSpz4*-silenced larvae. AMP gene expression levels in the fat body of *TmSpz4*-knock-down *T. molitor* larvae were assessed after injection with *E. coli* (Ec), *S. aureus* (Sa), or *C. albicans* (Ca). PBS was injected as a control 5 days post-*TmSpz4* silencing. At 24 h post-microbial challenge, the expression levels of several AMP genes, including those of *TmTen-1*, *TmTen-2*, *TmTen-3*, *TmTen-4*, *TmAtt-1a*, *TmAtt-1b*, *TmAtt-2*, *TmDef-1*, *TmDef-2*, *TmCol-1*, *TmCol-2*, *TmCec-2*, *TmTLP-1*, and *TmTLP-2*, were measured by qRT-PCR. ds*EGFP* was injected as a negative control, and *TmL27a* expression was measured as an internal control. All experiments were performed in triplicate. Asterisks indicate significant differences between ds*TmSpz4-* and ds*EGFP-*treated groups when compared by Student’s *t*-test (*p* < 0.05).

**Figure 7 ijms-21-01878-f007:**
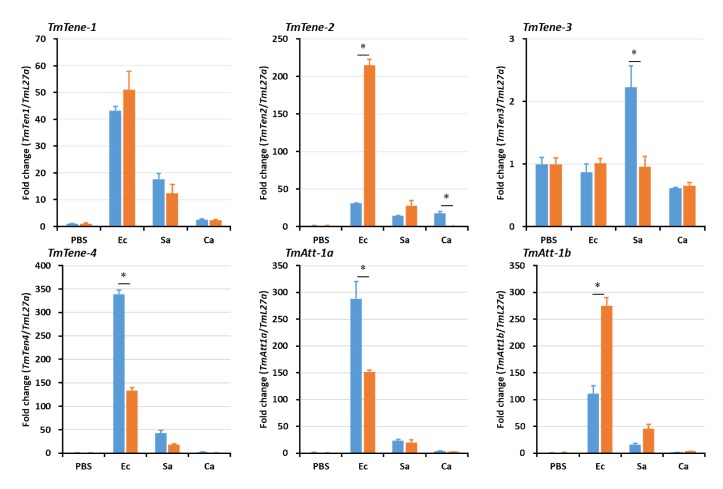
Antimicrobial peptide (AMP) induction patterns in gut of *TmSpz4*-silenced larvae. The antimicrobial peptide expression levels in *TmSpz4*-knock-down of *T. molitor* larval gut were performed by injecting either *E. coli* (Ec), *S. aureus* (Sa), or *C. albicans* (Ca). PBS was injected as a control 5 days post-*TmSpz4* silencing. At 24 h post-microbial challenge, the expression levels of several AMP genes, including those of *TmTen-1*, *TmTen-2*, *TmTen-3*, *TmTen-4*, *TmAtt-1a*, *TmAtt-1b*, *TmAtt-2*, *TmDef-1*, *TmDef-2*, *TmCol-1*, *TmCol-2*, *TmCec-2*, *TmTLP-1*, and *TmTLP-2*, were measured by qRT-PCR. ds*EGFP* was injected as a negative control, and *TmL27a* expression was measured as an internal control. All experiments were performed in triplicate. Asterisks indicate significant differences between ds*TmSpz4-* and ds*EGFP-*treated groups when compared by Student’s t-test (*p* < 0.05). *Effects of TmSpz4 on the expression patterns of NF-κB genes.*

**Figure 8 ijms-21-01878-f008:**
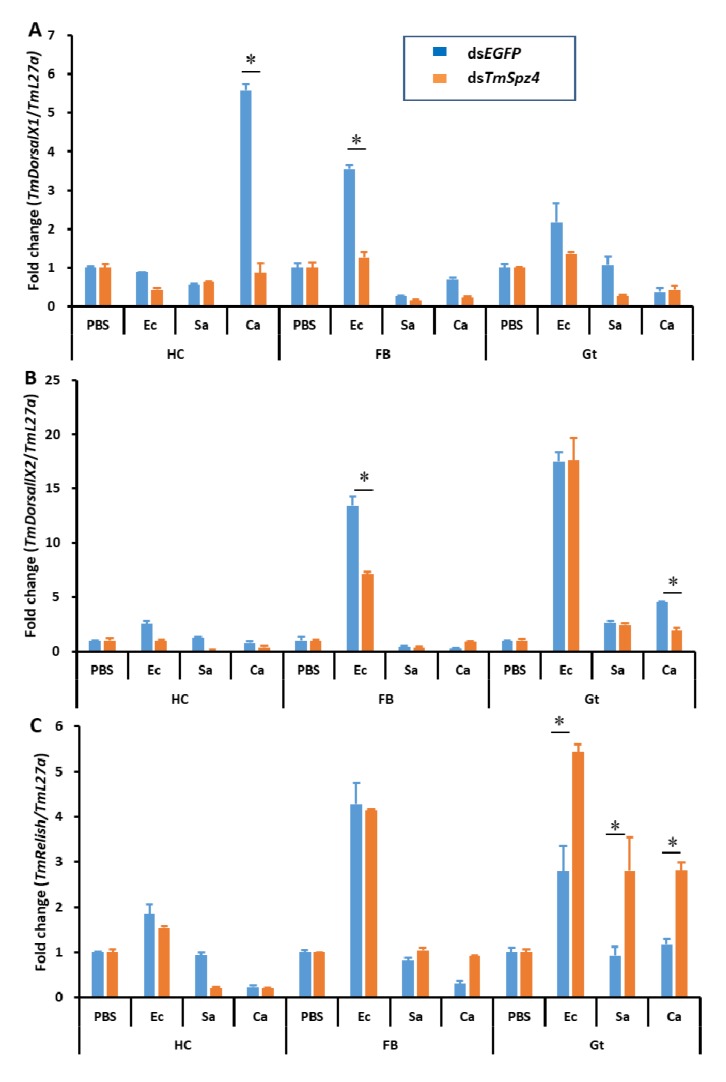
Effect of *Tmspz4* gene silencing on expression of NF-κB genes. Three different pathogens, *E. coli*, *S. aureus*, and *C. albicans*, were injected into *TmSpz4*-silenced *T*. *molitor* young larvae, and the expression of the NF-κB genes *TmDorX1* (**A**), *TmDorX2* (**B**), and *TmRel* (**C**) were then investigated by RT-qPCR. Larvae were injected with ds*EGFP* as a negative control, and *TmL27a* expression was assessed as an internal control. All experiments were performed in triplicate. Asterisks indicate significant differences in NF-κB gene expression between the ds*TmSpz4-* and *dsEGFP-*treated groups when compared by Student’s t-test (*p* < 0.05).

**Table 1 ijms-21-01878-t001:** Sequences of the primers used in this study.

Primer Name	Sequence (5′-3′)
*TmSpz4*-qPCR-Fw	GGCGATGCTCTTCCAGGAC
*TmSpz4*-qPCR-Rv	CGCGTTCACTCCTTTCATTTGG
*TmSpz4*-T7-Fw	TAATACGACTCACTATAGGGTCCAGATGTACTGTCGCGATG
*TmSpz4*-T7-Rv	TAATACGACTCACTATAGGGTTTCCTTCTGTACCAGTCGGG
*TmSpz4*-cloning-Fw	ACCGACACAACCACCAAAAG
*TmSpz4*-cloning-Rv	ATCCGTGATCTCCGGAAAAA
*TmSpz4*-cloning-FullORF-Fw	GACGGTGCCCACGAACTAT
*TmSpz4*-cloning-FullORF-Rv	AAGAGCACGAGCCTTGACAT
*TmL27a*_qPCR_Fw	TCATCCTGAAGGCAAAGCTCCAGT
*TmL27a*_qPCR_Rv	AGGTTGGTTAGGCAGGCACCTTTA
ds*EGFP*_Fw	TAATACGACTCACTATAGGGTCGTAAACGGCCACAAGTTC
ds*EGFP*_Rv	TAATACGACTCACTATAGGGT TGCTCAGGTAGTGTTGTCG
*TmTenecin-1*_Fw	CAGCTGAAGAAATCGAACAAGG
*TmTenecin-1*_Rv	CAGACCCTCTTTCCGTTACAGT
*TmTenecin-2*_Fw	CAGCAAAACGGAGGATGGTC
*TmTenecin-2*_Rv	CGTTGAAATCGTGATCTTGTCC
*TmTenecin-3*_Fw	GATTTGCTTGATTCTGGTGGTC
*TmTenecin-3*_Rv	CTGATGGCCTCCTAAATGTCC
*TmTenecin-4*_Fw	GGACATTGAAGATCCAGGAAAG
*TmTenecin-4*_Rv	CGGTGTTCCTTATGTAGAGCTG
*TmDefensin-1*_Fw	AAATCGAACAAGGCCAACAC
*TmDefencin-1*_Rv	GCAAATGCAGACCCTCTTTC
*TmDefencin-2*_Fw	GGGATGCCTCATGAAGATGTAG
*TmDefencin-2*_Rv	CCAATGCAAACACATTCGTC
*TmColoptericin-1*_Fw	GGACAGAATGGTGGATGGTC
*TmColoptericin-1*_Rv	CTCCAACATTCCAGGTAGGC
*TmColoptericin-2*_Fw	GGACGGTTCTGATCTTCTTGAT
*TmColoptericin-2*_Rv	CAGCTGTTTGTTTGTTCTCGTC
*TmAttacin-1a*_Fw	GAAACGAAATGGAAGGTGGA
*TmAttacin-1a*_Rv	TGCTTCGGCAGACAATACAG
*TmAttacin-1b*_Fw	GAGCTGTGAATGCAGGACAA
*TmAttacin-1b*_Rv	CCCTCTGATGAAACCTCCAA
*TmAttacin-2*_Fw	AACTGGGATATTCGCACGTC
*TmAttacin-2*_Rv	CCCTCCGAAATGTCTGTTGT
*TmCecropin-2*_Fw	TACTAGCAGCGCCAAAACCT
*TmCecropin-2*_Rv	CTGGAACATTAGGCGGAGAA
*TmThaumatin-likeprotein-1*_Fw	CTCAAAGGACACGCAGGACT
*TmThaumatin-like protein-1*_Rv	ACTTTGAGCTTCTCGGGACA
*TmThaumatin-like protein-2*_Fw	CCGTCTGGCTAGGAGTTCTG
*TmThaumatin-like protein-2*_Rv	ACTCCTCCAGCTCCGTTACA
*TmDorsal-X1*_qPCR_Fw	AGCGTTGAGGTTTCGGTATG
*TmDorsal-X1*_qPCR_Rv	TCTTTGGTGACGCAAGACAC
*TmDorsal-X2*_qPCR_Fw	ACACCCCCGAAATCACAAAC
*TmDorsal-X2*_qPCR_Rv	TTTCAGAGCGCCAGGTTTTG
*TmRelish*_qPCR_Fw	AGCGTCAAGTTGGAGCAGAT
*TmRelish*_qPCR_Rv	GTCCGGACCTCAAGTGT

Underline indicates T7 promoter sequences.

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
