# Peer review of "TmSpz4 Plays an Important Role in Regulating the Production of Antimicrobial Peptides in Response to Escherichia coli and Candida albicans Infections"

_ijms, 2020, doi:10.3390/ijms21051878_

Round 1

Reviewer 1 Report

In the manuscript entitled "TmSpz4 plays an important role in regulating the production of antimicrobial peptides in response to Escherichia coli and Candida albicans infections", Edosa et al. describe the gene spätzle 4 in Tenebrio molitor, and investigate its function in the insect immune system. Most of their work relies on qPCR experiments, with which they show that spätzle 4 is located in the main immune tissues and that the gene is expressed differentially overtime. Using RNAi, they show that when spätzle 4 is knocked down the insect is more susceptible to microbial infection, which is likely connected to the change in expression profiles of the antimicrobial peptides. In the end, they convincingly show that spätzle 4 is involved in the Toll signaling pathway regulation. Overall, they produced a large amount of data and they proceed rationally in presenting their results. Nevertheless, the paper presents in my opinion two major weaknesses.

The first one is the amateurish look of some figures. For instance Figure 4 is looking like a print screen from Excel. While it is not scientifically hurtful, it does not help to put all these nice results in a way that readers will be appreciating the work done. I am confident that the authors can improve greatly this point.

The second point, and maybe the more important, is that the Discussion is not very relevant. It consists on a lot of repetitions from the results with some attempt to put hypotheses that are not well supported by the results (why all of a sudden hormone regulation comes out of nowhere?). I would recommend to the authors to rebuild their discussion around the strength of their results and maybe to expand on some questioning the readers may have. I can think of, for instance:

- you state that there are 9 Spätzle genes in Tenebrio molitor. Why did you focus on the 4th and what could be the function of the others? Are they conserved enough that you knocked down all of them? There is not a single mention of the other genes in the discussion. I think that would be appropriate to discuss them.

- capitalize on what you discovered, for instance that you have a strong response of Toll signaling pathway against Gram negative. In the hegemony of the Drosophila model, that could be a relevant point to state that while the main pathways are retained, sensitivity to microbes may differ and so all what is described in Drosophila should not be taken as granted for insects, hence the need to develop other models for understanding insect immunity.

- you show that some AMP genes are upregulated in response to Spätzle 4 knock down when infected with E. coli (e.g. Attacin 1b), while the knock down also leads to a higher mortality. This seems contradictory. What could be the reason(s)? I think that is an important point to discuss and for which there is a vast literature to get ideas from.  

With an improved version, I think this manuscript can be published in IJMS.

Following are some more detailed and minor comments on the manuscript:

Abstract

You mention in the text "TmTencin". Correct me if I am wrong, because that is the first time I see those words, but did you mean "Tenecin"? Moreover, Tenecins are unrelated proteins that got their name because they were described in Tenebrio. Tenecin 1, 2 and 4 are inducible defensin, coleoptericin, and attacin, respectively, while Tenecin 3 is a Thaumatin. So, if I am not mistaken, you should correct to "Tenecin" and there is no need for "Tm" before, as the word itself relates to Tenebrio and is mostly used solely for Tenebrio.

l.39-40: that is just my opinion (and a question of style), but you say "principally" at the beginning of the sentence implying already that there are other factors beside the ones you have listed. So, I would get rid of "and other factors", because it does not bring anything besides stating that you did not want to list everything.

l.41: "germline-encoded" This expression is used mostly in vertebrate systems to state that, in opposite to adaptive immune system where antigen receptors are "encoded by genes that are assembled from individual gene segments during lymphocyte development", innate immunity uses receptors that "are encoded by intact genes inherited through the germline" (defined in "Immunobiology: The Immune System in Health and Disease. 5th edition."). So, in insects, there is no need for this expression as there is no adaptive immunity and this is just confusing to read.

l.45-57: these two paragraphs are highly redundant in the description of Spätzle. You could combine some of this information to make one paragraph, that would ease the reading.

l.58: there is 6 homologs and you mentioned "including spz-1", so your parentheses should read (Spz1-6).

l.79-80: Why TmSpätzle 4? Seems a little random without explaining here the interest on focusing on this one particularly. Especially after mentioning above that in Drosophila, Spz-1, -2, -5 and -7 (so no -4) promote AMPs production.

Figure 1: the signal peptide should be a sequence, so I assume that the red arrow marks the end of it. Maybe, you should mention it in the legend. As such, it seems that you imply that the red arrow is the peptide itself. Actually, a colored box would make more sense and would help identifying the beginning of the protein.

Figure 2: As you obtained your sequence from T. castaneum Spz4, which was itself annotated from Drosophila Spz4, that was quite expected to observe this phylogeny. I wonder the relevance of this figure in the paper. If you want to present it, maybe in supplemental material or you should expand the phylogeny to more Spätzle and more insects (why did you not put the 9 T. molitor Spätzle? It seems random to me that you put only -6 and -4). Nevertheless, as most of the "numbering" annotations were carried from one insect to another, again you should expect that the same numbers will cluster together. Apart if there is an actual relevance in this tree, I don't really get the point of such analysis.

l.100-113: I do not see any identification numbers for Tenebrio sequences. Did you submit them to any open access sequence library? That is important that the sequences you introduce in the paper for Tenebrio can be accessed easily.

Figure 3: you have no asterisks but you display differences with letters. Also, it would be good to put your statistical analyses in your legends (I assume you did an ANOVA with TukeyHSD post hoc test).

Figure 4: How did you proceed the normalization of infection results with PBS-injected larvae? You display error bars (which you do not explain what they are… SEM, SD?), so it means you did not do a mean/mean ratio. Infected larvae and PBS-injected ones are not paired, so if you randomly normalized 20 infected with 20 PBS-injected, the obtained results will be totally arbitrary and biased. In my opinion, you either put the PBS-injected as part of the graph, or you present fold changes from ratio mean infected/mean PBS-injected. Also, asterisks are not helping understanding your statistical analyses. Are they signifying differences between time points or between the different infection modalities? Or is it comparison between infected and PBS-injected? This figure is very confusing.

Figure 5: Did you check the other Spätzle genes? You stated that you had RNAseq data on the 8 others, so you could produce qPCR primers on them? By the way how closely related are they? Survival is usually displayed with Kaplan-Meier curves, because the death rate is punctual and not linear between days. What are the bars representing? (SEM, SD, 95% IC?). What statistical analysis did you use?

Also, that may sound like being picky, but you could have produced better looking figures as the different graphs are not consistent. The letters A, B, C and D are not placed on the same part, the y-label of 5D could have the (%) aligned like in 5C, x-label on 5D could be horizontal like the 3 other figures. These are small details for sure, but nice presented figures lead to better appreciation by readers, which can translate overall in a better recognition of your work. The choice of style is not what I'm scientifically evaluating, but I think it would help your work to improve this.

Figure 6: Considering the high number of tests you produced here, did you correct your p-values in some way? (maybe the software you used did it without you noticing? That would be nice to state if there was a correction in the t-tests).

Figure 7, 8 and 9 are called Figure 1, 2 and 3. Please correct.

Figure 8: As Figures 6, 7 and 8 are basically similar except for tissue location, I understand why legends 6 and 7 are similar, but why it is different for 8? Not that it is important, but it is just strange (different authors maybe?). I personally favor the version of Fig 6 and 7.

l.239: there is a space between TmDor X2 but not for TmDorX1. Also, in Figure 9, TmDorX1 is named TmDor-1 with the addition of a hyphen and there is again a space for TmDor -X2. Try to be consistent overall. I can assure you that it makes the reading harder as it seems that all those genes are different while they are the same.

l.272: typo, should be "Investigation"

l.278-280: your conclusion is kind of a stretch. You associate things without any strong support. You could speculate and propose that it should be investigated, but certainly not that your data suggest that there is a hormone-based regulation of Spätzle.

l.281-296: This paragraph is badly written. I don't follow most of the rational in there. I guess you simply want to state that while it was described in Drosophila that IMD was more specific to Gram negative bacteria, your results are here showing that the Toll pathway respond to them as well, which is supported by other results. Nothing novel, but I agree that it is a relevant discussion point. Try to state simpler what you meant, because this paragraph is too complex for what it aims to state.

l.297-307: I feel that this paragraph could be improved by stating what novelties your results bring in the current knowledge, which is the regulation of AMPs under TmSpz4. Most of what you state is basically description of other results disconnected to your own results.

l.313: typo "DDorsal" should be "Dorsal". Also, "Dorsal and Relish" not "or", you did both.

l.334: put capital G for Gram.

l.341-342: italics for E. coli, S. aureus and C. albicans.

l.344: you say BLASTn here but stated tblastn on l.84. Which one is it? (BLASTn is actually simply BLAST, so correct if this one is the correct one). You mention you used the gene, so I assume you used BLAST and not tBLASTn.

l.359: but no deposition of sequence anywhere? You should absolutely put the sequence somewhere.

l.366: if you refer to the protein, I think you should write TmSPZ4.

l.369: you did a maximum likelihood analysis for the phylogeny, you mentioned it in the results, but it should be in your method as well.

l.372: you mention that you extracted egg RNA. However, nowhere in the manuscript you mention results on eggs. What happen to those results?

l.422: italics for T. molitor.

l.428: your survival experiments are paired (dsEGFP vs dsTmSpz4). So, why did you do a Tukey's multiple test? This is a post hoc test, fit for when you have >2 modalities. Usually survival statistical analyses like the one your present (categorical) are based on a Log-rank test.

l.452: this paragraph is redundant with paragraph 4.7 for the survival part. An ANOVA is completely inappropriate for survival analysis as the dependent variable is binary (dead or alive). I am also confused on how you input your external control with the 2^(-ddCt) method. This could be only done if you had paired data, which is not your case, as PBS-injected insects are different from infected insects. You can find this method when for instance you have sample from a same individual (for example blood of a mouse before and after a treatment). If you are confident you used a fit method for what you state, I would appreciate an expanded explanation. Also, you have a dedicated Data analysis paragraph, but you do not display all the statistical analyses you did in it. Considering that some analyses don't even have an explanation, I think you should rewrite this part.

Reviewer 2 Report

In this study, Edosa et al have characterized the role of TmSpZ4 protein of Tenebrio molitor in immune response to E. coli and C. albicans. The authors have documented that TmSpz4 is inducible upon infections and silencing its expression leads to decreased survival of T. molitor. The authors have also shown some data implicating the role of the NFkB pathway in this process. This paper is well written and the results have been clearly represented. The study lacks novelty since the role of Spatzle proteins in immune response has been previously implicated and the same group have published a very similar study on TmSpz6.

Following are some minor revisions that need to be done prior to acceptance:

Abbreviations when used for first time in the manuscript needs to be spelled out. IMD line #43, UPR line #70 and so on. The authors need to Cite their Tmspz6 paper since its published and especially since the current study design is very similar to their Tmspz6 publication. In line #100, the figure legend reads that the phylogenetic analysis is that of TmSpz6. Is this Tmspz6 or Tmspz4? Figure numbers seem to be incorrect. After Figure 6, the figures are labelled Fig 1-3 again. Is this supplemental figures? If so where is figure 7,8 and 9 which they refer to in the body of the paper? The authors need to correct these mistakes. Most of the data in the manuscript is that of mRNA levels as opposed to protein levels. Is there a reason to showing mRNA levels of NFKB mediators instead of doing a western blot looking at the activation status of these signaling proteins? The authors need to explain/discuss why some AMP levels decrease while some increase upon siRNA knockdown of TmSpz4. Is there differential regulation of AMPs?

Round 2

Reviewer 1 Report

I think the authors have greatly improved their manuscript and delivered a nice article. They have very deeply answered my comments with relevant and accurate responses. Their work is now quite suitable for being published.